# Influence of health literacy on acceptance of influenza and pertussis vaccinations: a cross-sectional study among Spanish pregnant women

Enrique Castro-Sánchez,[1] Rafael Vila-Candel,[2,3] Francisco J Soriano-Vidal,[3,4,5,6] Esther Navarro-Illana,[3] Javier Díez-Domingo[3,5]

EC-S and RV-C contributed equally.

[1]NIHR Health Protection Research Unit (HPRU) in Healthcare Associated Infections (HCAI) and Antimicrobial Resistance (AMR), Imperial College London, London, UK
[2]Department of Obstetrics and Gynaecology, Hospital Universitario de la Ribera, Valencia, Spain
[3]Faculty of Nursing, Universidad Católica de Valencia 'San Vicente Mártir', Valencia, Spain
[4]Xàtiva-Ontinyent Health Department, Xàtiva, Spain
[5]Foundation for the Promotion of Health and Biomedical Research in the Valencian Region (FISABIO), Valencia, Spain
[6]Department of Nursing, University of Alicante. San Vicente del Raspeig, Alicante, Spain

**Correspondence to**
Dr Enrique Castro-Sánchez;
e.castro-sanchez@imperial.ac.uk

## ABSTRACT

**Objectives** Immunisations against influenza and *Bordetella pertussis* infection are recommended to pregnant women in Valencia (Spain), yet vaccination rates remain low. Health literacy (HL) appears as a crucial factor in vaccination decision-making. We explored the relation between HL of pregnant women and decisions to receive influenza and pertussis immunisations.

**Setting** University hospital in Valencia (Spain).

**Participants** 119 women who gave birth at a hospital in Valencia (Spain) between November 2015 and May 2016. Women in the immediate postpartum period (more than 27 weeks of gestation), between November 2015 and May 2016 were included in the study. Women with impairments, language barriers or illiteracy which prevented completion of the questionnaires, or those who were under 18 years were excluded from enrolment.

**Primary and secondary outcome measures** HL level; influenza and pertussis immunisation rate; reasons for rejection of vaccination.

**Results** 119 participants were included (mean age 32.3±5.5 years, 52% primiparous, 95% full-term deliveries). A higher education level was associated with Short Assessment of Health Literacy for Spanish Adults _50 (adjusted $R^2$=0.22, p=0.014) and Newest Vital Sign (adjusted $R^2$=0.258, p=0.001) scores. Depending on the scale, 56%–85% of participants had adequate HL. 52% (62/119) and 94% (112/119) of women received influenza and pertussis immunisation, respectively. Women rejecting influenza vaccine had a higher HL level (measured by SALHSA_50 tool) than those accepting it (Kruskal-Wallis test p=0.022). 24% of women who declined influenza vaccination felt the vaccine was unnecessary, and 23% claimed to have insufficient information.

**Conclusions** Influenza vaccination rate was suboptimal in our study. Women with high HL were more likely to decline immunisation. Information from professionals needs to match patients' HL levels to reduce negative perceptions of vaccination.

## BACKGROUND

Despite its benefits, influenza vaccine coverage among pregnant women remains low.[1] Some determinants associated with

### Strengths and limitations of this study

► Validated health literacy screening tools were administered to pregnant women to identify health literacy levels. Immunisation status was obtained from official vaccination records.
► Screening tools used in the study have been validated in Spanish-speaking populations in the USA but not Spain.
► Further research could focus on the development and use of pregnancy-specific scales.

vaccination rejection include insufficient information by professionals and underestimation of infection risks during pregnancy.[2–4]

However, pregnancy is a risk factor for severe influenza, a main reason for hospital admission during gestation.[5] The administration of influenza vaccine to pregnant women would protect immunised mothers and infants. As the safety of the vaccine is well established, its administration is recommended during any trimester of gestation. Globally, influenza vaccination coverage is uneven, ranging from 15%–43% in Europe,[6] to 50% in the USA.[7] In Spain, there are no published data on national influenza vaccination coverage among pregnant women; however, our review in 2014–2015 reported vaccination rates of 40.5% in pregnant women in our health department.[8]

Vaccination against *Bordetella pertussis* is equally recommended to all pregnant women in Valencia (Spain) since January 2015 due to outbreaks of whooping cough.[9] Women are offered immunisation on the third trimester, ideally between weeks 27 and 36 of gestation.[6] As with influenza, maternal immunisation also benefits newborns.[10] According to WHO, 195 000 children under 5 years died in 2008 of whooping cough. More than 80% of deaths occurred in children younger than 6

months of age. The number of whooping cough cases has increased since 2011 worldwide, including the European Union, and among children and young adults. In Spain, the case incidence has shifted from 739 cases in 2008 to 3088 cases in 2011, a global rate of 6.73/100 000 habitants/year for that year. Additionally, eight deaths in 2001 were attributed to whooping cough.[11] Of concern, there are currently no published data regarding whooping cough vaccination coverage among pregnant women in Spain. However, reports on the incidence of whooping cough in 2015 are available, indicating 17.99 cases per 100 000 people, with provisional data for 2016 suggesting a marked decline in reported cases.[12]

Among the factors determining vaccination acceptance, health literacy (HL) refers to the knowledge and skills required when making health decisions.[13] Essential HL skills include reading, writing, numeracy and searching for information.[14 15] Inadequate HL has been associated with poor health outcomes including inadequate self-caring and preventive behaviours such as vaccination.[16] Standardised tools for assessing HL are available, yet mostly in English[17] and focused on US society. European researchers have developed questionnaires,[18] and some tools (Short Assessment of Health Literacy for Spanish Adults; SAHLSA_50),[19 20]Newest Vital Sign (NVS)[21–24] and Single Item Literacy Screener (SILS)[25] have been validated in Spanish language but not for Spanish citizens.

Although vaccination is especially relevant for pregnant women and wider public health,[26] no studies have been conducted in Spain exploring the relationship between HL and vaccine acceptance. We hypothesise that pregnant women with limited HL may be less likely to accept influenza and pertussis vaccinations in Valencia (Spain).

## METHODS
### Study population and sampling criteria
We conducted a cross-sectional study in women who had given birth at La Ribera University Hospital (Hospital Universitario de La Ribera, HULR) in Valencia (Spain). The HULR serves a population of 250 000 citizens and is the only hospital providing maternity services to pregnant women in the area, with an annual average of 1600 births in the year when the study was carried out. The influenza and pertussis vaccine policy in the HULR mirrors the national policy, where vaccines are offered systematically, by community midwives and family doctors, to all women free of charge. In 2015, the influenza vaccination rate for the whole Valencian Community was 34.4%.

Immunisation campaign in Spain starts in October and concludes in March. In order to avoid seasonality, we included all women during the study period. Women in their immediate postpartum period (more than 27 weeks of gestation), between November 2015 and May 2016 were included in the study. We excluded women with impairments, language barriers or illiteracy. Illiterate women were excluded from the study due to their

inability to complete the HL screening tools which were self-administered. Any help from the researchers would likely influence the results.[27] Women younger than 18 years were also excluded from taking part. Prior to data collection, written consent was obtained from each participant.

For recruitment, we systematically approached all women admitted to the maternity ward, every 4 days. To calculate the sample size, we used the SALHSA_50 tool as a reference with a cut-off score of 0–37 for inadequate literacy. Accepting an alpha risk of 0.05 and a beta risk of 0.2 in a bilateral contrast, with a common SD of 7.0[28] and a loss to follow-up rate of 10%, we estimated that 102 participants would be required.

### Measurements
During the immediate postpartum (24–48 hours after delivery), we collected sociodemographic, obstetric variables and vaccination status through review of medical records, and HL from each woman through interview with the researcher in charge.

Participants' HL was determined using three screening tools:

1. SALHSA_50: evaluates word recognition and reading comprehension through a 50-item tool. Quantitative scores classify individuals with 'adequate' (score: 38–50 points) or 'inadequate' HL (score: 0–37 points). The tool has been validated for Hispanics in the USA.
2. NVS: evaluates reading and numeracy through six questions about the label of an ice-cream. The sum score (0–6 points) categorises individuals with high likelihood of limited literacy (score: 0–1 points), possibility of limited literacy (score: 2–3 points) and adequate literacy (score: 4–6 points). It has been validated for the Hispanic population in the USA.[23] It has high sensitivity, but it can misclassify people with adequate HL.[29]
3. SILS: it asks patients how often they need help when reading health instructions. The response is recorded on a 5-point Likert-type scale (1-never, 2-rarely, 3-sometimes, 4-often and 5-always) and categorised as adequate or inadequate. Scores greater than 2 indicate some difficulty with reading materials.[25]

Regarding vaccination, we analysed: (1) influenza or pertussis vaccination status during pregnancy; (2) if vaccinated, health centre where vaccinated; (3) which healthcare provider recommended it and (4) if vaccination rejection, reasons for declining. Vaccination status was corroborated using the regional vaccination registry which records all vaccines received by patients.[8]

Other variables collected through review of medical records included: age, country of origin, civil status, occupation, education, gestational age, parity, type of delivery, risk factors during pregnancy (without risk or low risk, pregestational or gestational diabetes, thyroid pathology, pre-eclampsia, twin pregnancy and assisted reproduction treatment).

## Statistical analysis

In the univariate analysis, quantitative variables were described with means and SD or median and IQR, depending on the normality of their distribution. The Kolmogorov-Smirnov goodness-of-fit test was used to determine the normality of distributions. In the bivariate analysis, the $\chi^2$ test was used between the qualitative variables and the vaccination status. To compare the medical risk factors during pregnancy related to vaccination, OR with a 95% CI was calculated.

The non-parametric Mann-Whitney U test was used when the normality hypothesis was rejected when comparing independent samples with the categorised values of NVS and SAHLSA_50 and vaccination acceptance. To identify the variables explaining the level of HL according to each screening tool, a series of multivariate analyses were conducted. The multivariate lineal regression analysis (Wald statistic) was used regarding the explanatory covariates for the quantitative tools, NVS and SALHSA_50, and a multinomial model was constructed for the qualitative scale SILS. The level of statistical significance was set at 0.05. SPSS for Windows V.22.0 (IBM) was used for data analysis.

## Patient and public involvement

Patients were not involved in the development of the research questions, the design of the study or the recruitment of participants. Aggregated study results will be published on the website of the hospital, in suitable language.

## RESULTS

Out of a total of 168 women who initially consented to be included in the study, 49 were excluded (29%) for the following reasons: 10 (20%) were breast feeding, 16 (33%) had language barriers, 16 (33%) were busy, 4 (8%) were absent from their room and 3 (6%) were unwell. Therefore, the study sample comprised 119 participants (71%).

Table 1 presents the sociodemographic characteristics of participants. The mean age was 32.3±5.5 years, with 29.5±5.4 as mean age for the first pregnancy. Fifty-two per cent (62) were primiparous. The mean gestational age at delivery was 39.1±1.5, with 95% (113) full-term deliveries (37–42 weeks).

The information and recommendation about vaccination came mainly from their midwives (94%), in 4% from the family doctor and 2% of women did not provide any information. As we wanted to be as sure as possible of the vaccination status of each participant, we validated the vaccination status reported by the participants with the immunisation status recorded in the official electronic immunisation registry. We corroborated that all women without immunisation recorded on the electronic record had not been vaccinated.

Regarding HL screening tools, the correlation between SAHLSA_50 and SILS was moderate, inversely proportional and significant (r=−0.251, p=0.007). The correlation between NVS and SAHLSA_50 was moderate and significant (r=0.349, p<0.001). The correlation between NVS and SILS was moderate, inversely proportional and also significant (r=−0.307, p=0.001).

We also analysed the influence of participants' education on HL level and the scales of assessment. Higher education was directly related to higher SAHLSA_50 (r=0.244, p<0.001) and NVS (r=0.366, p=0.002) scores. This relationship, however, was not present in the SILS scale.

## Vaccination status

Seventeen per cent (20/62) of women had been vaccinated against influenza prior to pregnancy. Gestational influenza vaccination coverage was 52% (62/119). The vaccine was administered to 5% (4/62) of women by week 20, and to 16% (10/62) in the last weeks of gestation (more than 36 weeks). Concerning pertussis vaccine, 94% (112/119) of women had it during pregnancy, with 86% (96/112) vaccinated between weeks 27 and 32 of gestation. All women vaccinated against influenza were simultaneously vaccinated against whooping cough. There were no significant differences in sociodemographic or obstetric characteristics between pregnant vaccination status for influenza or pertussis (p=0.15 and p=0.35, respectively) (data not shown).

The reasons for rejection of women who were not vaccinated against influenza during pregnancy (57) are shown in figure 1. Twenty-five per cent (14/57) felt that the vaccine was unnecessary, 23% (13/57) claimed to have received insufficient information from health professionals and 14% (8/57) claimed that they had never been infected. The reasons reported by women declining vaccination against pertussis were lack of information from health professionals (4/7; 57%) and lack of any prenatal care (3/7; 43%).

## Health literacy

In the NVS scale, we obtained an average score of 3.7±1.6 with values between 0 and 6. These scores were categorised as inadequate (13% (16/119)), limited (30% (36/119)) and adequate HL (56% (67)). SAHLSA_50 scores were 44.1±4.4 out of 50. Eighty-six per cent (102/119) of women had adequate HL levels (SAHLSA-50 score >37). According to the SILS, 24% (29/119) women replied 'never' needing help when reading information, 29% (35/119) 'rarely', 27% (32/119) 'sometimes' and only 6% (7/119) replied 'often' and 13% (16/119) replied 'always'.

To identify variables explaining HL levels according to each screening tool, multivariate analyses were conducted. Multivariate lineal regression was used regarding the explanatory covariates for quantitative tools, NVS and SALHSA_50. For these, the level of education was found to be statistically significant (NVS (adjusted $R^2$=0.258; p=0.001) and SALHSA_50 (adjusted $R^2$=0.220; p=0.014)). A multinomial model was constructed for the

**Table 1** Sociodemographic, clinical and obstetric characteristics of the sample by influenza vaccine status (n=119)

| | Total row | Unvaccinated, n=57 | | Vaccinated, n=62 | | P values* |
|---|---|---|---|---|---|---|
| | N | N | % | N | % | |
| **Civil status** | | | | | | |
| With partner | 48 | 20 | 35 | 28 | 45 | 0.458 |
| Married/civil partner | 67 | 35 | 61 | 32 | 52 | |
| Separated/divorced | 4 | 2 | 4 | 2 | 3 | |
| **Level of education** | | | | | | |
| Primary school | 40 | 20 | 36 | 20 | 32 | 0.296 |
| Secondary school | 42 | 19 | 33 | 23 | 37 | |
| University | 37 | 18 | 44 | 19 | 31 | |
| **Employment status** | | | | | | |
| I | 13 | 9 | 16 | 4 | 6 | 0.083 |
| II | 66 | 35 | 61 | 31 | 50 | |
| III | 2 | 0 | 0 | 2 | 3 | |
| IV | 1 | 0 | 0 | 1 | 2 | |
| V | 37 | 13 | 23 | 24 | 39 | |
| **Country of Origin** | | | | | | |
| Spain | 104 | 51 | 89 | 53 | 85 | 0.261 |
| Another EU country | 8 | 5 | 9 | 3 | 5 | |
| Non-EU country | 1 | 0 | 0 | 1 | 2 | |
| Central-South America | 6 | 1 | 2 | 5 | 8 | |
| **Pertussis vaccine** | | | | | | |
| Unvaccinated | 7 | 7 | 12 | 0 | 0 | 0.269 |
| Vaccinated | 112 | 50 | 88 | 62 | 100 | |
| **Medical risk factors during pregnancy** | | | | | | |
| None/low risk | 92 | 45 | 79 | 47 | 76 | 0.570 |
| Pregestational/gestational diabetes | 7 | 2 | 3 | 5 | 8 | |
| Thyroid pathology | 7 | 5 | 9 | 2 | 3 | |
| Pre-eclampsia | 1 | 0 | 0 | 1 | 2 | |
| Twin pregnancy | 3 | 1 | 2 | 2 | 3 | |
| ART | 9 | 4 | 7 | 5 | 8 | |
| **NVS categories** | | | | | | |
| Inadequate (0–1 points) | 13 | 6 | 10 | 7 | 11 | 0.219 |
| Limited (2–3 points) | 38 | 14 | 25 | 24 | 39 | |
| Adequate (4–6 points) | 68 | 37 | 65 | 31 | 50 | |
| **SAHLSA categories** | | | | | | |
| Inadequate (0–37 points) | 17 | 6 | 10 | 11 | 18 | 0.261 |
| Adequate (38–50 points) | 102 | 51 | 89 | 51 | 82 | |
| **SILS categories** | | | | | | |
| Never | 29 | 13 | 23 | 16 | 26 | 0.947 |
| Rarely | 34 | 17 | 30 | 17 | 27 | |
| Sometimes | 33 | 17 | 30 | 16 | 26 | |
| Often | 8 | 4 | 7 | 4 | 6 | |
| Always | 15 | 6 | 10 | 9 | 14 | |

I, self-employed, higher professional or managerial employment; II, employee; III, student;  IV,  stay-at-home mother; V, unemployed.
*$X^2$.
ART, assisted-reproduction treatment; EU, European Union; NVS, Newest Vital Sign; SAHLSA, Short Assessment of Health Literacy for Spanish Adults; SILS, Single Item Literacy Screener.

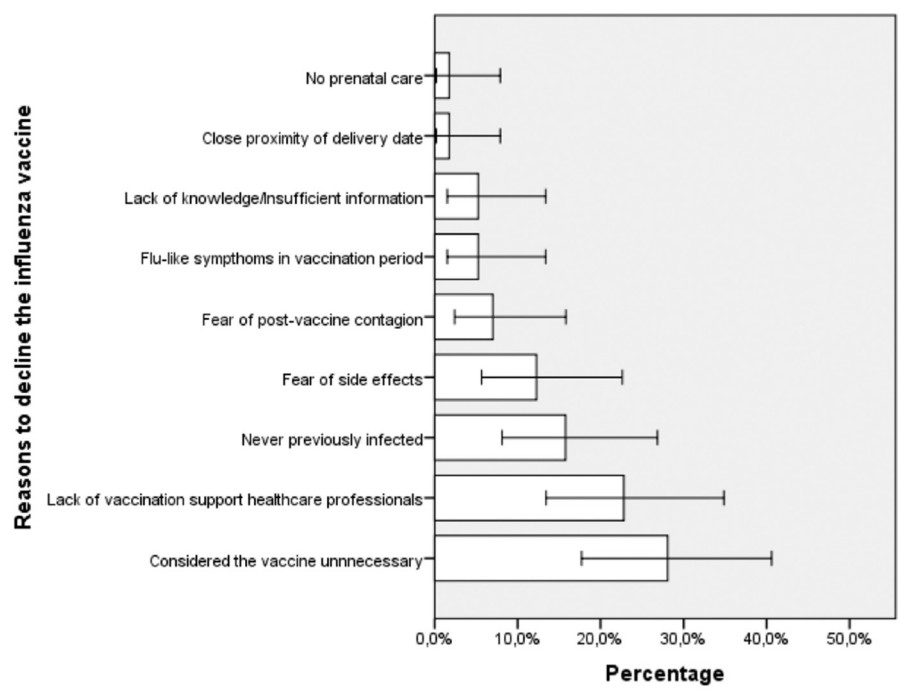

**Figure 1** Reasons given by participants to decline influenza vaccination.

qualitative scale SILS, observing no statistically significant differences.

There were no differences in NVS and SILS scores between women who declined and those who accepted influenza vaccination (Mann-Whitney U test, p=0.320 and p=0.942, respectively). However, for SAHLSA_50 (median=44.5; IQR=5.0 vs 45.0; IQR=5.5) the differences were statistically significant (Mann-Whitney U test, p=0.019) (figure 2).

Later, scores from the quantitative HL screening tools (NVS, SALHSA_50) were distributed in quartiles (figure 3). For the NVS scale, we found no statistically significant different between women who had accepted

or declined vaccination (p=0.532). However, such difference was seen when using the SALHSA_50 tool (Kruskal-Wallis test, p=0.022). The median number of women vaccinated in the bottom quartile was 8 (95% CI 7.0 to 9.0) versus 24 (95% CI 23.0 to 25.0) in the top quartile.

We were interested in examining the characteristics of the women who were excluded from the study (49). We conducted an analysis of missing values for the three HL screening tools using the multiple imputation chained equations method.[30] Again, for the NVS scale, we found no statistically significant difference between women who had accepted or declined vaccination (p=0.372)

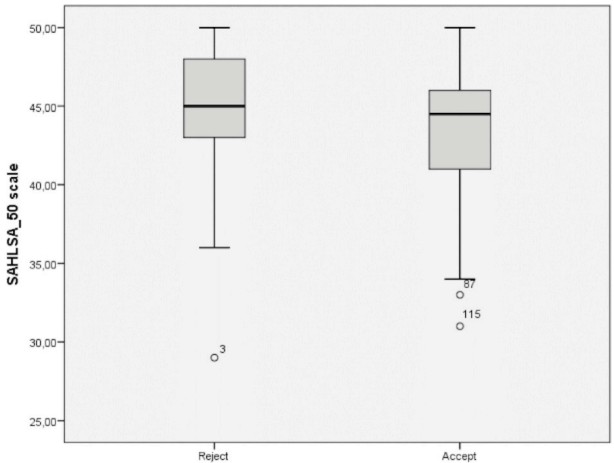

**Figure 2** Relationship between acceptance of influenza vaccination and SAHLSA_50 scale (n=119). SAHLSA, Short Assessment of Health Literacy for Spanish Adults.

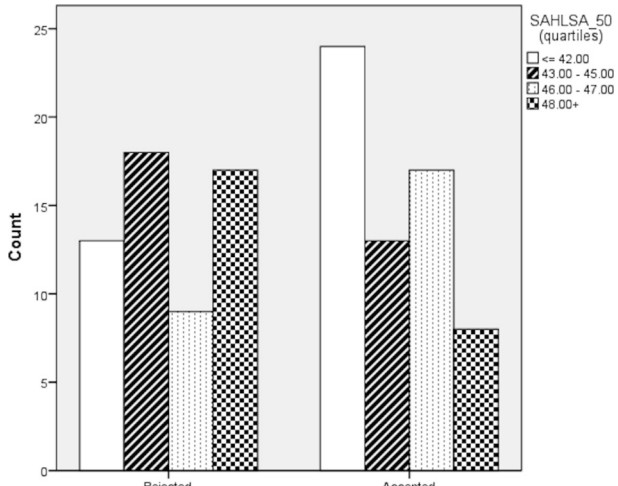

**Figure 3** Relationship between acceptance of influenza vaccination and SAHLSA_50 scale distribution by quartiles (n=119). SAHLSA, Short Assessment of Health Literacy for Spanish Adults.

and, instead, such difference was seen when using the SALHSA_50 tool (Kruskal-Wallis test, p=0.003). The median number of women vaccinated in the bottom quartile was 11 (95% CI 9.0 to 12.0) vs 28 (95% CI 27.0 to 29.0) in the top quartile.

Regarding the NVS, scores between pertussis-vaccinated and unvaccinated women were similar (median=4.0; IQR=0.0 vs median=4.0; IQR=2.75), like the SAHLSA_50 scale (median=45.0; IQR=0.0 vs median=45.0; IQR=5.0). We also did not find any difference with the results from the SILS tool.

## DISCUSSION

Vaccination is an essential public health intervention. We focused on pregnant women, an especially vulnerable population, and studied the acceptance of two vaccines underused in our community.[8] Few studies evaluating HL and vaccination have been conducted thus far and, up to now, none had focused on pregnant women.

In our study, influenza vaccination did not reach recommended levels although coverage slightly exceeded Australian[31] but not US rates.[32] In Valencia, coverage has progressively improved from 2011 (8.5%) to 2015 (34.4%).[8] Regarding pertussis, the 97% vaccination rate improves on Belgian (39%)[33] or UK (70%)[6] rates. The disparity between pertussis and influenza immunisation rates has not been previously addressed in detail.[6] We believe that in our setting, fear to pertussis—perhaps influenced by mass media[34] and fuelled by the increasing number of cases—could explain such high vaccination prevalence. Indeed, the pertussis vaccination programme was commenced following a surge in the number of cases and deaths. Clinicians may have therefore been keener to ensure that pregnant women got vaccinated and may have framed their advice more assertively. On the other hand, the disinterest from health professionals in providing information about influenza vaccination together with maternal perceptions that influenza vaccine was unnecessary were the most frequently cited causes of vaccine rejection, in agreement with prior studies.[5–8 35] This position obviously ignores the benefits of acquired immunity for the newborn which could reduce perinatal infections.[36]

We found that NVS classified 58% of participants with adequate HL. However, this figure increased up to 89% if SAHLSA_50 was used. Currently, there are no publications comparing both scales simultaneously in the same population. Such discrepancy between screening tools could be of much relevance as, of the tools pragmatically chosen for our research, only SALHSA_50 was predictive of vaccination in pregnant women. However, women with high SALHSA_50 scores were more likely to decline influenza vaccination, perhaps due to preconceived ideas; it might also be that women with high HL have more abilities to look for information on the internet or other sources and construct a narrative that supports such preconceptions, leading to declining this vaccination.[37] Such narratives would also not be challenged if

professionals fail to adequately inform them or focus their persuasion solely on rational, data-based reasons instead of complementing such evidence with other emotional and behavioural aspects.[36 38 39] These results diverge from current evidence[4] in this group of women possibly highly involved in their healthcare, as already explored.[40]

As perhaps expected, HL screening results were directly related to the education of participants and thus, a higher level of education was associated with higher HL. Interestingly, other authors have reported that a higher level of education is associated with higher rates of vaccine rejection and hesitation.[35 41] In fact, it would appear that the emerging relationship between HL and vaccination described by those authors may be represented as an 'inverted U' shape' (ie, high and low HL levels equally associated with low vaccination).

The analysis of missing values would help resolve some of the challenges originated from the incomplete responses. If cases with missing data were to be systematically different to cases with complete information, then results could be equivocal.[30] In our case, however, the analysis of missing values did not produce different results to the original analysis conducted without imputed values.

Our study presents limitations. Although there are approximately 51 HL tools available,[17] experiences in Spain with these instruments have been few and limited to the Health Literacy Survey—European Union[42] or the eHealth Literacy[43] tools. In addition, none of these tools have been validated in Spain, yet they have been so in Spanish-speaking US populations. Moreover, as there are no scales specifically focused on pregnant women, our questionnaire selection was eminently pragmatic and based on ease of use (SILS), robustness (SAHLSA_50) and reliability (NVS). Additionally, the routine use of HL screening tools in clinical practice remains nevertheless controversial, as such routine screening has shown no benefits yet could have undesirable effects for patients.[44]

Decisions related to vaccination may be influenced by the information provided, the communication approaches and attitudes of by health professionals.[35 36] Since there is currently no standardised approach to determine the abilities that pregnant women have to make effective use of the information provided, we hypothesise that information offered to each woman will be more or less similar and, therefore, women with low HL may be more likely to make suboptimal decisions because of such deficit. Logically, this does not consider efforts that professionals may make to compensate for any difficulties in understanding. Although exploring such efforts was outside the remit of our work, it would be interesting to investigate this aspect in future studies, together with any supporting materials used by professionals.

## CONCLUSION

Vaccination is an essential public health measure, and pregnant women can particularly benefit from this intervention. Identifying determinants of vaccination such

as HL would facilitate an adequate use of resources to encourage shared decision-making, ultimately resulting in optimal vaccination rates. Our findings suggesting a relation between high HL and rejection of vaccination encourage further research to identify and describe the factors involved in such relation and implement mitigating initiatives.

**Acknowledgements** The authors are grateful to the Universidad Católica de Valencia and to the HULR for their support and participation in this project, and to all the women who have also taken part. The authors also thank Professor Daniel Lee for authorising the use of the SALHSA_50 tool in their research.

**Contributors** EC-S and RV-C conceptualised and designed the study. RV-C and FS-V collected data. EC-S, RV-C and FS-V carried out the data analyses. EC-S, RV-C, EN-I, JDD and FS-V drafted the initial manuscript. EC-S, RV-C, FS-V, EN-I and JDD reviewed and revised the manuscript. All authors read and approved the final manuscript.

**Funding** This work was supported by the Universidad Católica de Valencia "San Vicente Mártir" (Spain) through a competitive grant call [PRUCV/2015/639]. ECS is affiliated with the National Institute for Health Research (NIHR) Health Protection Research Unit (HPRU) in Healthcare Associated Infection and Antimicrobial Resistance at Imperial College London in partnership with Public Health England (PHE), and the NIHR Imperial Patient Safety Translational Research Centre. ECS has received a Wellcome ISSF Faculty Fellowship at Imperial College London, an Early Career Research Fellowship from the Antimicrobial Research Collaborative at Imperial College London, and acknowledges the support of the Florence Nightingale Foundation.

**Disclaimer** The views expressed are those of the authors and not necessarily those of the NHS, the NIHR, the Department of Health, or Public Health England. The funders had no role in study design; collection, analysis and interpretation of data; writing the report; and the decision to submit the report for publication.

**Competing interests** None declared.

**Patient consent** Not required.

**Ethics approval** The study was conducted in compliance with the Declaration of Helsinki and it was approved by the Research Ethics and Research Committee of Hospital Universitario de La Ribera on 10/07/15.

**Provenance and peer review** Not commissioned; externally peer reviewed.

**Data sharing statement** No additional data are available.

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
