## [Reviewer comments · BMJ Open]

ARTICLE DETAILS

TITLE (PROVISIONAL)	Influence of health literacy on acceptance of influenza and pertussis vaccinations: a cross-sectional study among Spanish pregnant women
AUTHORS	Castro-Sánchez, Enrique; Vila-Candel, Rafael; Soriano-Vidal, Francisco; Navarro-Illana, Esther; Domingo, Javier Diez

VERSION 1 – REVIEW

REVIEWER	Corinne Vandermeulen KU Leuven, Belgium
REVIEW RETURNED	26-Dec-2017

GENERAL COMMENTS	General comments: This manuscript concerns a study in which the association between health literacy is studied in relation to getting vaccinated during pregnancy. The hypothesis was that low health literacy might be related to less acceptance of vaccination during pregnancy. Detailed comments Abstract - Line 48-50: the word “reject” is used twice which does not fit. One of the two should be accepting.- It would be good to add how many participants were vaccinated for influenza and pertussis for the entire group. Introduction - The introduction would benefit from some more background on the recommendation in Spain regarding vaccination of pregnant women and of the background burden of disease of pertussis in infants (how many infants have died of pertussis in the past years, has there been an increase before 2015. There was a decline after 2016, what are the main reasons for this decline,...). Material and Methods - Study participants – exclusion criteria: women with impairments, language barriers and general illiteracy were excluded from the study. It seems to me that, when researching health literacy, these women should have been included as they were part of the intended population. Given the research hypothesis, excluding these women from the study will bias the results. I understand that women being illiterate cannot read, but they should have been included in the study as being health illiterate.- The use of the SILS questionnaire seems questionable as the results are likely to be subjective and influenced by the purpose of the study.
--

	Results  - General results: of the women who were excluded, 20 women refused to participate: why did they refuse to participate: was this noted? If this was because they refused vaccination during pregnancy, this might also bias the results. - Table 1: please add a column with the total numbers per row. Andfor NVS and SAHLSA categories add the cut-off for HL. - Health Literacy (p12): it is said that there is a relationship between the level of education and ???: it is not clear whether this is a relationship with HL or with vaccination status. Also the direction of the influence is not indicated (i.e. negative or positive) Discussion  - In general it seems that you have conflicting results with the different tests. This is, however, not been discussed in detail in your discussion. - p13: you state that the two vaccines are underutilized in your community, but do you have any reference coverage rates to back this up. - p14: Belgian vaccination rates have been updated by data published by Maertens et al. Vaccine 2016. - p14: middle paragraph: you state that women with high SALHSA_50 scores were more likely to decline influenza vaccination due to pre-conceived ideas. It might also be that women with high HL have more abilities to look for information on the internet and are deceived by the information they found and as such decline vaccination. There is a good publication by Stahl et al (2016, Médecine et maladies Infectieuses. The impact of the web and social networks on vaccination. New challengesand opportunities offered to fight against vaccine hesitancy.) in which this is clearly explained. - p14: last paragraph: do you have any indication in your study to confirm or deny that the opposite of your primary hypothesis is true? Table 1? No statistically significant difference, but trend? - One of the limitations is your limited number of women participating in your study, even though according to your power calculations this might be enough. I believe that 119 women is not enough, especially if you exclude an important part of your target population. Conclusion  - You state that your results will facilitate an adequate use of resources to encourage shared decision-making. How will you implement this? How will gynaecologists and midwives know which women are HL and which are not? Which message do they need to give to either of them?
--	--

REVIEWER	Yuelian Sun The Department of Clinical Epidemiology, Aarhus University, Denmark
REVIEW RETURNED	02-Jan-2018

GENERAL COMMENTS	The authors explored the relation between health literacy (HL) of pregnant women and decisions to receive influenza and pertussis immunizations, which is an interesting and important topic. The main concern of the study is that they collected the information including health literacy postpartum and the participants in the final analyses were selected and may not representative for the population in the served area.
--

	1. Nearly one third of the study population (49/168) were excluded from the analyses. Are their vaccination status same as those in the final analyses? How about other factors that authors could get from medical report? Are those who declined in the study more likely to have an eventful pregnancy outcome? Are participants in the analyses representative? Would the reasons of declining vaccination differ among mothers with an eventful or uneventful pregnancy? 2. Is HULR the only hospital providing birth delivery service to pregnant women in the area? If no, is the population the hospital covers selected to some extent, for example those with higher education or income status? Is there any delivery at home? 3. Why did the authors choose Nov 2015 to May 2016 as the study period? Can season be a factor for vaccination rate? Are the distribution of vaccinated and non-vaccinated pregnant women same according to season? 4. The authors claimed that the aim of the study was to explore the relation between health literacy (HL) of pregnant women and decisions to receive influenza and pertussis immunizations. Since this is a cross-sectional study and information of the health literacy was collected after the birth of the child, it is hard to say that the health literacy is the main factor for the vaccination. For example the authors asked information from health professionals only among those rejected to take the influenza vaccination. How about the status of knowledge from the health professionals among the participants who accepted the influenza vaccination? Is that the main factor that affect them accepting the vaccination? 5. The article applied three instruments to measure the health literacy and the findings between health literacy and vaccination are inconsistent according to the three measurements. Which one can better assess the health literacy and which findings we should believe in on the association between health literacy and vaccination rate? 6. Although the vaccination rate for pertussis vaccine is high (94%), it will be good to collect and present data on important factors like 'information from health professionals on the pertussis vaccine' or media report, which may be the main factor that affect the vaccination rate between the influenza vaccination and pertussis vaccination rather than health literacy. 7. The authors stated that the collect vaccination status both from each woman (p7, line 31) and vaccination registry (p8, line 19). Is the information from the two sources consistent? 8. The cut-off score of inadequate literacy is not consistently presented in the text (p7 line 16 and line 40). 9. Please specify information from interview and medical reports separately (p8, line 24-33). 10. The sentence 'Women who were not vaccinated against influenza during pregnancy (57) were asked about their reasons for rejection' (p11, line 51-52) sounds to belong to the method section. The message from the figure 1 and the numbers are inconsistent at least for the group who 'felt that the vaccine was unnecessary'. The bar message is around 28% while the number shows 25%? In the
--	---

	legend of figure 1, please provide the number of participants. 11. How did the author get the common standard deviation of 7.0 (p7, line 18)?
--	---

VERSION 1 – AUTHOR RESPONSE

Reviewers' Comments to Author:

Reviewer: 1

Reviewer Name: Corinne Vandermeulen

Institution and Country: KU Leuven, Belgium Competing Interests: None declared

General comments:

This manuscript concerns a study in which the association between health literacy is studied in relation to getting vaccinated during pregnancy.

The hypothesis was that low health literacy might be related to less acceptance of vaccination during pregnancy.

Detailed comments

Abstract

- It would be good to add how many participants were vaccinated for influenza and pertussis for the entire group.:

Influenza (62 vaccinated/57 non vaccinated), pertussis (112 vaccinated/7 non vaccinated).

Introduction

- The introduction would benefit from some more background on the recommendation in Spain regarding vaccination of pregnant women and of the background burden of disease of pertussis in infants (how many infants have died of pertussis in the past years, has there been an increase before 2015. There was a decline after 2016, what are the main reasons for this decline,...):

Thank you. According to WHO, 195.000 children under 5-years died in 2008 of whooping cough. More than 80% of deaths occur in those younger than 6 months of age. The number of whooping cough cases has increased since 2011 worldwide, including the European Union, and among children and young adults. In Spain, case incidence since 2011 has shifted from 739 cases in 2008 to 3.088 cases in 2011, a global rate of 6.73/100,000 hab/year for 2011. In 2011 there were 8 whooping cough deaths in Spain (World Health Organization. Pertussis vaccines: WHO Position paper. Wkly Epidemiol Rec. 2010;85:385-400). In 2014, the rate increased to 9.7/100,000 hab/year, and particularly among 0-4 years-age (rate 51.1/100,000 hab/year).

Material and Methods

- Study participants – exclusion criteria: women with impairments, language barriers and general illiteracy were excluded from the study. It seems to me that, when researching health literacy, these women should have been included as they were part of the intended population. Given the research hypothesis, excluding these women from the study will bias the results. I understand that women being illiterate cannot read, but they should have been included in the study as being health illiterate.:

We respectfully disagree with the reviewer on 2 points here; first, illiterate women were excluded from the study due to their inability to complete the complete the health literacy screening tools, which were self-administered. Any help from the researchers would likely influence the results (Davis RE, Couper MP, Janz NK, Caldwell CH, Resnicow K. Interviewer effects in public health surveys. Health Education Research. 2010;25(1):14-26.). Secondly, it is perfectly possible to be illiterate yet to be adequately health literate. Individuals could receive information orally, or using pictograms, and be able to make effective health decisions.

- The use of the SILS questionnaire seems questionable as the results are likely to be subjective and influenced by the purpose of the study. :

We cannot see how the reviewer is able to justify such statement. The SILS tool, as included in the manuscript, has been validated adequately.

Results

- General results: of the women who were excluded, 20 women refused to participate: why did they refuse to participate: was this noted? If this was because they refused vaccination during pregnancy, this might also bias the results.:

As in any research, characteristics of individuals who decline to participate may be different to those enrolled in the study. However, we would not be able to ascertain the reasons for refusal to participate in the study (finding that out would, effectively, make them participants). Following from that, we would not be able to ascertain the vaccination status of the individuals, as collecting or seeking such information without their consent would be unethical.

- Health Literacy (p12): it is said that there is a relationship between the level of education and ????: it is not clear whether this is a relationship with HL or with vaccination status. Also the direction of the influence is not indicated (i.e. negative or positive):

Our study found education to be positively associated with the level of health literacy. Such relation was statistically significant. This association seems logical. We included a paragraph in the discussion where we make explicit that we are talking about education and health literacy, rather than education and vaccination.

Discussion

- In general it seems that you have conflicting results with the different tests. This is, however, not been discussed in detail in your discussion.:

We cannot see where the conflict lies; in our manuscript we report how different screening tests may allocate individuals to different health literacy strata. This issue is highlighted in our manuscript when reporting about the different screening tools used.

- p13: you state that the two vaccines are underutilized in your community, but do you have any reference coverage rates to back this up

Vila-Candel R, Navarro-Illana P, Navarro-Illana E, et al. Determinants of seasonal influenza vaccination in pregnant women in Valencia, Spain. BMC Public Health. 2016. doi:10.1186/s12889-016-3823-1.

- p14: Belgian vaccination rates have been updated by data published by Maertens et al. Vaccine 2016.

Thank you, if the reviewer refers to Maertens K, Caboré RN, Huygen K, Hens N, Van Damme P, Leuridan E Pertussis vaccination during pregnancy in Belgium: Results of a prospective controlled cohort study. Vaccine. 2016 Jan 2;34(1):142-50. doi: 0.1016/j.vaccine.2015.10.100, this paper only offers information about whooping cough cases among children ,rather than reporting on immunisation among pregnant women.

- p14: middle paragraph: you state that women with high SALHSA_50 scores were more likely to decline influenza vaccination due to pre-conceived ideas. It might also be that women with high HL have more abilities to look for information on the internet and are deceived by the information they found and as such decline vaccination. There is a good publication by Stahl et al (2016, Médecine et maladies Infectieuses. The impact of the web and social networks on vaccination. New challenges and opportunities offered to fight against vaccine hesitancy.) in which this is clearly explained

- p14: last paragraph: do you have any indication in your study to confirm or deny that the opposite of your primary hypothesis is true? Table 1? No statistically significant difference, but trend?

:

We report on lines 18-25, pg 13, and Figure 3.

- One of the limitations is your limited number of women participating in your study, even though according to your power calculations this might be enough. I believe that 119 women is not enough, especially if you exclude an important part of your target population. : **This opinion is disappointing. We offered a sample size calculation and would expect the reviewer to focus on the merits of such calculation, rather than a belief...**

Conclusion

- You state that your results will facilitate an adequate use of resources to encourage shared decision-making. How will you implement this? How will gynecologists and midwives know which women are HL and which are not? Which message do they need to give to either of them? :

The implementation of our findings would be a different scenario to the experience reported in our manuscript. The use of point-of-care screening tests such as SAHLSA-50 may offer opportunities to clinicians to identify women with low health literacy, or perhaps even better, by recognizing that an important proportion of women may have low health literacy, clinicians and organizations could design services and offer clinical practice that would be of benefit to all users, regardless of their health literacy levels.

Reviewer: 2

Reviewer Name: Yuelian Sun

Institution and Country: The Department of Clinical Epidemiology, Aarhus University, Denmark

Competing Interests: None declared

The authors explored the relation between health literacy (HL) of pregnant women and decisions to receive influenza and pertussis immunizations, which is an interesting and important topic. The main concern of the study is that they collected the information including health literacy postpartum and the participants in the final analyses were selected and may not representative for the population in the served area. :

As we conducted a sample size calculation, we cannot see what else would be needed to ensure representativeness...

1. Nearly one third of the study population (49/168) were excluded from the analyses. Are their vaccination status same as those in the final analyses? :

48% vs 52%.

How about other factors that authors could get from medical report?

Our study focused on health literacy and associated variables, rather than all variables that have been suggested to influence decisions about vaccination in pregnant women.

Are those who declined in the study more likely to have an eventful pregnancy outcome?

As mentioned to the previous reviewer, it would not be appropriate to examine data from individuals not enrolled in the study. Additionally, we are not sure what 'eventful pregnancy outcome refers to...

Are participants in the analyses representative?

According to our simple size calculation, 112 participants were required and we enrolled 119 individuals in the study.

Would the reasons of declining vaccination differ among mothers with an eventful or uneventful pregnancy?

We do not really understand what the reviewer means by 'eventful or uneventful pregnancy, unfortunately.

2. Is HULR the only hospital providing birth delivery service to pregnant women in the area?

Yes.

If no, is the population the hospital covers selected to some extent, for example those with higher education or income status? Is there any delivery at home?

We are not sure why would it be relevant to consider home births, which are extremely infrequent in Spain and, by definition, would mean that women are not seen in our hospital service. Our study focused on women who gave birth at hospital, and therefore seems reasonable to ignore other modes of delivery.

3. Why did the authors choose Nov 2015 to May 2016 as the study period?

Immisation campaign in Spain starts in October and concludes in March. Precisely, in order to avoid seasonality, we included all women during the study period. Can season be a factor for vaccination rate? Are the distribution of vaccinated and non-vaccinated pregnant women same according to season? Season may be a factor to explain vaccination rate, but it would be likely to affect both groups (women with low and high health literacy) equally.

4. The authors claimed that the aim of the study was to explore the relation between health literacy (HL) of pregnant women and decisions to receive influenza and pertussis immunizations. Since this is a cross-sectional study and information of the health literacy was collected after the birth of the child, it is hard to say that the health literacy is the main factor for the vaccination.

We did not claim that health literacy was the main factor determining vaccination, but one of the factors, as reported in other studies in different settings, populations and immunisations. We did not identify statistically significant differences in the sociodemographic and obstetric characteristics of women vaccinated vs not vaccinated. Such lack of statistically significant differences were seen for both flu (p=0,15) or pertussis (p=0,35) vaccinations. We later on explored the relation between vaccination and the different study variables, identifying statistically significant differences between women who had accepted flu vaccination before pregnancy vs those who had not accepted vaccination

For example the authors asked information from health professionals only among those rejected to take the influenza vaccination. How about the status of knowledge from the health professionals among the participants who accepted the influenza vaccination? Is that the main factor that affect them accepting the vaccination?

Such hypothesis, whilst plausible, was not the focus of our current study. However, in a our previous publication (Vila-Candel R, Navarro-Illana P, Navarro-Illana E, et al. Determinants of seasonal influenza vaccination in pregnant women in Valencia, Spain. BMC Public Health. 2016. doi:10.1186/s12889-016-3823-1.) we identified that “The information and recommendation of vaccination came mainly from their midwives (89%), in 9% (9/100)from the family doctor and 2% of women did not provide any information”

5. The article applied three instruments to measure the health literacy and the findings between health literacy and vaccination are inconsistent according to the three measurements. Which one can better assess the health literacy and which findings we should believe in on the association between health literacy and vaccination rate?

Thank you- we conducted an analysis to determine if the results from the different scales were correlated (pg 11, lines 2-12). Our study did not aim to establish whether any of the scales was better or worse than any other, and our Discussion highlighted patient classification issues related to the use of screening tools. We focused on exploring if SALHSA results were associated with vaccination status.

6. Although the vaccination rate for pertussis vaccine is high (94%), it will be good to collect and present data on important factors like ‘information from health professionals on the pertussis vaccine’ or media report, which may be the main factor that affect the vaccination rate between the influenza vaccination and pertussis vaccination rather than health literacy.

It would seem plausible and even obvious that if healthcare professional recommendations for influenza immunization were not effective to induce vaccination, then they also would be unlikely to have any effect on pertussis vaccination. We included a reference in our discussion regarding the impact of news and media on vaccination with pertussis. We however did not suggest that health literacy was an explanatory factor regarding the difference in vaccination rates seen between the 2 types of immunisations.

7. The authors stated that the collect vaccination status both from each woman (p7, line 31) and vaccination registry (p8, line 19). Is the information from the two sources consistent?

As we wanted to be as sure as possible of the vaccination status of each participant, we added a layer of validation to women’s self-reported status by corroborating such status with the immunization status recorded in the official electronic record. In a previous study (Vila-Candel R, Navarro-Illana P, Navarro-Illana E, et al. Determinants of seasonal influenza vaccination in pregnant women in Valencia, Spain. BMC Public Health. 2016. doi:10.1186/s12889-016-3823-1.) we determined that all women with lack of immunisation recorded on the electronic immunisation record had not received the vaccine.

8. The cut-off score of inadequate literacy is not consistently presented in the text (p7 line 16 and line 40).

We have to disagree with the reviewer. We report “from 0-37” and “<37”.

10. The message from the figure 1 and the numbers are inconsistent at least for the group who 'felt that the vaccine was unnecessary'. The bar message is around 28% while the number shows 25%? In the legend of figure 1, please provide the number of participants.

We are grateful for this, we will review what could be our mistake in the graph.

11. How did the author get the common standard deviation of 7.0 (p7, line 18)?

Padilla-Santoyo P, Vílchez-Román C. Psychometric properties of the SAHLSA-50, a standardized test to evaluate the health literacy. Rev Per Obst Enf 2008; 4:90–95.

VERSION 2 – REVIEW

REVIEWER	Corinne Vandermeulen KU Leuven, Belgium
REVIEW RETURNED	12-Feb-2018

GENERAL COMMENTS	The authors replied correctly to all raised comments of the initial review.
---

REVIEWER	Yuelian Sun Department of Clinical Epidemiology, Aarhus University Hospital, Denmark
REVIEW RETURNED	08-Mar-2018

GENERAL COMMENTS	Reviewers' Comments to Author: Reviewer: 2 14) The authors explored the relation between health literacy (HL) of pregnant women and decisions to receive influenza and pertussis immunizations, which is an interesting and important topic. The main concern of the study is that they collected the information including health literacy postpartum and the participants in the final analyses were selected and may not representative for the population in the served area. : In the revised version of the manuscript we have further clarified the implications of the number of participants with missing values and the effect it could have on our results. We have additionally included now an analysis using imputation of missing values techniques, which suggest however that our results are unlikely to be affected by those missing values. Comments: Could the authors provide information on how many women gave births in the study period in the hospital and the vaccination rate for the whole population? Concerning the sample calculation, is it calculated to explore the association between HL and the influenza vaccination? Should it be dependent on the vaccination rate? I guess the sample should be larger to explore the association between HL and the pertussis vaccination. In the background, the authors added more information on pertussis vaccination while the results had very limited information about association between HL and pertussis vaccination. It seems that HL did not play any role in pertussis vaccination. It reported that 'The reasons reported by women declining vaccination against pertussis were lack of information from health professionals (4 [57%]) and lack of any prenatal care (3 [43%])'. Since most texts on the association between HL and vaccination is for influenza vaccination, the author used 'vaccination' referred to
--

	'influenza vaccination', which should be specified. 15) Nearly one third of the study population (49/168) were excluded from the analyses. Are their vaccination status same as those in the final analyses? : Thank you for this reflection- the results were fairly similar, 48% vs 52%. Comments: Fine. 16) How about other factors that authors could get from medical report? We agree with the reviewer that there are multiple factors that are known to influence vaccination decisions in pregnant women. However, our study focused solely on health literacy and associated variables, rather than all variables that have been suggested to influence decisions about vaccination in pregnant women. Comments: It is interesting to explore why HL have different association between influenza vaccination and pertussis vaccination. Besides HL, there must be other explanations. Could the authors add more in the discussion? 17) Are those who declined in the study more likely to have an eventful pregnancy outcome? Many thanks for highlighting this issue- as mentioned in response #5 to the other reviewer, it would be inappropriate to examine data from individuals not enrolled in the study. Additionally, we are unsure about what 'eventful pregnancy outcome' means. Comments: I mean if those who declined in the study were more likely to have a baby with congenital malformation, preterm birth, low Apgar score, or infection during pregnancy, etc, which made the mother declining to participate in the study? 18) Are participants in the analyses representative? According to our simple size calculation, 112 participants were required and we enrolled 119 individuals in the study. Please consider the additional analysis now included in the manuscript and discussed in response #12. Comments: Please refer to my comment about sample calculation in 14). 19) Would the reasons of declining vaccination differ among mothers with an eventful or uneventful pregnancy? As above in response #17, we do not really understand what the reviewer means by 'eventful or uneventful pregnancy', unfortunately. Comments: Please refer to my comment about sample calculation in 17). 20. Is HULR the only hospital providing birth delivery service to pregnant women in the area? Yes. Comments: Fine. 21) If no, is the population the hospital covers selected to some
--	--

	extent, for example those with higher education or income status? Is there any delivery at home? This is an interesting suggestion. However, we are not sure why considering home births would be relevant- they are extremely infrequent in Spain and, by definition, would mean that women are not seen in our hospital service. Our study focused on women who gave birth at hospital, and therefore seems reasonable to ignore other modes of delivery on this occasion. 22) Why did the authors choose Nov 2015 to May 2016 as the study period? Immunisation campaign in Spain starts in October and concludes in March. Precisely, in order to avoid seasonality, we included all women during the study period. Can season be a factor for vaccination rate? Are the distribution of vaccinated and non-vaccinated pregnant women same according to season? Season may be a factor to explain vaccination rate, but it would be likely to affect both groups (women with low and high health literacy) equally. Comments: Fine. 23) The authors claimed that the aim of the study was to explore the relation between health literacy (HL) of pregnant women and decisions to receive influenza and pertussis immunizations. Since this is a cross-sectional study and information of the health literacy was collected after the birth of the child, it is hard to say that the health literacy is the main factor for the vaccination. We would like to take this opportunity to clarify the purpose of our study to the reviewers. We did not claim that health literacy was the main factor determining vaccination or vaccination decisions, but one of the factors, as reported by other studies in different settings, focused on different populations and immunisations. We did not identify statistically significant differences in the sociodemographic and obstetric characteristics of women vaccinated vs not vaccinated. Such lack of statistically significant differences was seen for both influenza ($p=0,15$) or pertussis ($p=0,35$) vaccinations. We later on explored the relation between vaccination and the different study variables, identifying statistically significant differences between women who had accepted flu vaccination before pregnancy vs those who had not accepted vaccination. Comments: The author's reply did not answer the question on type of the study. If we assume the authors design a prospective cohort study and assess the HL before or at the early of pregnancy, would the results be similar to the current design? 24) For example the authors asked information from health professionals only among those rejected to take the influenza vaccination. How about the status of knowledge from the health professionals among the participants who accepted the influenza vaccination? Is that the main factor that affect them accepting the vaccination? We agree with the reviewer on the important influence that healthcare workers can have to support immunisation decisions among pregnant women. Such hypothesis, whilst plausible, was not the focus of our current study. Our group has touched upon this on a previous publication (Vila-Candel R, Navarro-Illana P, Navarro-Illana E, et al. Determinants of seasonal influenza vaccination in pregnant
--	---

	women in Valencia, Spain. BMC Public Health. 2016. doi:10.1186/s12889-016-3823-1.) were we identified that "The information and recommendation of vaccination came mainly from their midwives (89%), in 9% (9/100) from the family doctor and 2% of women did not provide any information" Comments: Thank you for the information. 25) The article applied three instruments to measure the health literacy and the findings between health literacy and vaccination are inconsistent according to the three measurements. Which one can better assess the health literacy and which findings we should believe in on the association between health literacy and vaccination rate? We are grateful to the reviewer for considering this matter. We conducted an analysis to determine whether the results from different scales correlate (pg 11, lines 2-12). Our study did not aim to establish whether any of the scales was better or worse than any other, and our Discussion highlighted patient classification issues related to the use of these, and other, screening tools. We focused on exploring if SALHSA results were associated with vaccination status. Comments: As the authors stated in the discussion 'the routine use of HL screening tools remains nevertheless controversial, as routine screening has shown no benefits yet could have undesirable effects.' What message can clinician take home from this study? 26) Although the vaccination rate for pertussis vaccine is high (94%), it will be good to collect and present data on important factors like 'information from health professionals on the pertussis vaccine' or media report, which may be the main factor that affect the vaccination rate between the influenza vaccination and pertussis vaccination rather than health literacy. It would seem plausible that, if healthcare professional recommendations for influenza immunization were not effective to induce influenza vaccination decisions, then they would also be unlikely to have any effect on pertussis vaccination. We included a reference in our Discussion regarding the impact of news and media on vaccination with pertussis. We however did not suggest that health literacy was an explanatory factor regarding the difference in vaccination rates seen between the 2 types of immunisations. Comments: Could the authors provide plausible explanations regarding the difference in vaccination rates seen between the 2 types of immunisations? 27. The authors stated that the collect vaccination status both from each woman (p7, line 31) and vaccination registry (p8, line 19). Is the information from the two sources consistent? Many thanks for raising this important point. We wanted to be as sure as possible of the vaccination status of each participant. To achieve that certainty, we added another layer of validation to the status reported by women and corroborated such status with the immunization information recorded in the official electronic immunization registry. In a previous study (Vila-Candel R, Navarro-Illana P, Navarro-Illana E, et al. Determinants of seasonal influenza vaccination in pregnant women in Valencia, Spain. BMC Public Health. 2016. doi:10.1186/s12889-016-3823-1.), we determined that all women who lacked immunisation recorded on the electronic
--	---

	immunisation record had indeed not received the vaccine. Comments: Ok. 28. The cut-off score of inadequate literacy is not consistently presented in the text (p7 line 16 and line 40). We have to respectfully disagree with the reviewer. In our manuscript we report "from 0-37" and "<37" as cut-off scores. comments: In page 7 the authors stated that 'Quantitative scores classify individuals with "adequate" (score: 37-50 points) or "inadequate" HL (score: 0-37 points).' Which group should a woman be if she got a score of 37, adequate or inadequate? I think the way reported in table 1 is a right way to do (adequate 38-50). 29. The message from the figure 1 and the numbers are inconsistent at least for the group who 'felt that the vaccine was unnecessary'. The bar message is around 28% while the number shows 25%? In the legend of figure 1, please provide the number of participants. We are grateful for this, we have now reviewed the figure and rectified the mistake in the graph. Comments: Ok. 30. How did the author get the common standard deviation of 7.0 (p7, line 18)? Many thanks- we used the information from Padilla-Santoyo P, Vilchez-Román C. Psychometric properties of the SAHLSA-50, a standardized test to evaluate the health literacy. Rev Per Obst Enf 2008; 4:90-95. Comments: Ok.
--	---

VERSION 2 – AUTHOR RESPONSE

Reviewers' Comments to Author:

Editorial Requests:

- Please revise your title so that it includes your study design. This is the preferred format for the journal

Thanks. We have no included the study design in the title.

- Please re-write the 'strengths and limitations' section on page 4. It shouldn't be a summary of the study and its findings. As a reminder, this section should contain up to five short bullet points, no longer than one sentence each, that relate specifically to the methods or design of the study reported (see: <http://bmjopen.bmj.com/site/about/guidelines.xhtml#articletypes>).

Many apologies, the section has been now re-written to the Section appropriately.

- Please remove the 'what is already known' and 'what this study adds' sections (these are not requirements for BMJ Open).

Done, many apologies.

- Along with your revised manuscript, please provide a completed copy of the STROBE checklist (<http://www.strobe-statement.org/>).

We have now included the STROBE checklist.

Reviewers' Comments to Author:

Reviewer: 1

The authors replied correctly to all raised comments of the initial review.

We are grateful to the reviewer for their help and input.

Reviewer: 2

14) The authors explored the relation between health literacy (HL) of pregnant women and decisions to receive influenza and pertussis immunizations, which is an interesting and important topic. The main concern of the study is that they collected the information including health literacy postpartum and the participants in the final analyses were selected and may not representative for the population in the served area. :

In the revised version of the manuscript we have further clarified the implications of the number of participants with missing values and the effect it could have on our results. We have additionally included now an analysis using imputation of missing values techniques, which suggest however that our results are unlikely to be affected by those missing values.

Comments:

Could the authors provide information on how many women gave births in the study period in the hospital and the vaccination rate for the whole population?

We provide further information. There were 930 deliveries during the study period, and the vaccination rate for pregnant women in the period 2015-6 was 36.4%. We have added this information on pages 6-7.

Concerning the sample calculation, is it calculated to explore the association between HL and the influenza vaccination? Should it be dependent on the vaccination rate? I guess the sample should be larger to explore the association between HL and the pertussis vaccination.

Many thanks for this reflection. As we did not know the level of health literacy in our population (dependent variable), we estimated the sample size via the comparison of averages for independent groups assuming unknown yet comparable variances between groups, selecting 2 groups of equal size. The SD suggested by the literature regarding the SALHSA_50 was 7 points. Thus, for a 95% CI, 80% power, and 10% estimated losses, the required sample size for each group should comprise 51 participants.

In the background, the authors added more information on pertussis vaccination while the results had very limited information about association between HL and pertussis vaccination. It seems that HL did not play any role in pertussis vaccination. It reported that 'The reasons reported by women declining vaccination against pertussis were lack of information from

health professionals (4 [57%]) and lack of any prenatal care (3 [43%])'. Since most texts on the association between HL and vaccination is for influenza vaccination, the author used 'vaccination' referred to 'influenza vaccination', which should be specified.

We agree with the reviewer that the available evidence regarding HL and pertussis is modest. We had not provided enough epidemiological information about pertussis in a previous draft of the manuscript. We updated the information following a request from a reviewer. We strived to present a balanced overview of the results obtained in our study regarding both vaccinations, and we feel that, overall, the paper bundles the findings when appropriate yet focuses on key vaccine-specific when necessary. The Discussion, as it stands, is a good example of such considerations.

16) It is interesting to explore why HL have different association between influenza vaccination and pertussis vaccination. Besides HL, there must be other explanations. Could the authors add more in the discussion?

Many thanks, in the Discussion we had hypothesized that perhaps the surge in number of cases including deaths from pertussis infection may have been responsible for the large difference in number of vaccinated women, compared with influenza. We had highlighted the role that mass media may have played in heightening perceptions of pertussis as a much more pressing or lethal issue than influenza. We have further expanded the Discussion to suggest that clinicians may have therefore been more assertive framing their vaccination messages regarding pertussis compared with influenza, which may have been seen as a recurring health problem.

17) Are those who declined in the study more likely to have an eventful pregnancy outcome?

Many thanks for highlighting this issue- as mentioned in response #5 to the other reviewer, it would be inappropriate to examine data from individuals not enrolled in the study. Additionally, we are unsure about what 'eventful pregnancy outcome' means.

Comments:

I mean if those who declined in the study were more likely to have a baby with congenital malformation, preterm birth, low Apgar score, or infection during pregnancy, etc, which made the mother declining to participate in the study?

We are grateful to the reviewer for this clarification. We did not collect information about the outcomes of the pregnancy, so we cannot really provide any information on this aspect. It could be argued that women with lower health literacy may end up experiencing worse perinatal outcomes due to worse or more untimely access to healthcare, as seen in other clinical situations or healthcare issues.

18) Are participants in the analyses representative?

According to our simple size calculation, 112 participants were required and we enrolled 119 individuals in the study. Please consider the additional analysis now included in the manuscript and discussed in response #12.

Comments:

Please refer to my comment about sample calculation in 14).

Thank you, we have clarified this aspect in the answer to 14)

19) Would the reasons of declining vaccination differ among mothers with an eventful or uneventful pregnancy?

As above in response #17, we do not really understand what the reviewer means by 'eventful or uneventful pregnancy', unfortunately.

Comments:

Please refer to my comment about sample calculation in 17).

Thank you. As mentioned, we did not collect information about pregnancy outcomes. In our study, however, the reasons provided by women in our study seemed to focus on external factors (lack of information from professionals) and indeed perceptions about lack of negative effects from influenza infection.

23) The authors claimed that the aim of the study was to explore the relation between health literacy (HL) of pregnant women and decisions to receive influenza and pertussis immunizations. Since this is a cross-sectional study and information of the health literacy was collected after the birth of the child, it is hard to say that the health literacy is the main factor for the vaccination.

We would like to take this opportunity to clarify the purpose of our study to the reviewers. We did not claim that health literacy was the main factor determining vaccination or vaccination decisions, but one of the factors, as reported by other studies in different settings, focused on different populations and immunisations. We did not identify statistically significant differences in the sociodemographic and obstetric characteristics of women vaccinated vs not vaccinated. Such lack of statistically significant differences was seen for both influenza ($p=0,15$) or pertussis ($p=0,35$) vaccinations. We later on explored the relation between vaccination and the different study variables, identifying statistically significant differences between women who had accepted flu vaccination before pregnancy vs those who had not accepted vaccination.

Comments:

The author's reply did not answer the question on type of the study. If we assume the authors design a prospective cohort study and assess the HL before or at the early of pregnancy, would the results be similar to the current design?

Many apologies for not providing a clear response. In essence, as we conducted a cross-sectional study we are not able to establish a causal relation between the variable and the outcome of interest. With such study design it is not possible to exclude other factors as responsible for the results obtained. Within the timespan of a pregnancy it is unlikely that women's health literacy levels would have been significantly modified by the interactions with health care workers or the information provided by them, and therefore a prospective study may offer similar results. Finally, our study replicates multitude of previous others focused on health literacy and a clinical outcome of interest.

25) The article applied three instruments to measure the health literacy and the findings between health literacy and vaccination are inconsistent according to the three measurements. Which one can better assess the health literacy and which findings we should believe in on the association between health literacy and vaccination rate?

We are grateful to the reviewer for considering this matter. We conducted an analysis to determine whether the results from different scales correlate (pg 11, lines 2-12). Our study did not aim to establish whether any of the scales was better or worse than any other, and our Discussion highlighted patient classification issues related to the use of these, and other, screening tools. We focused on exploring if SALHSA results were associated with vaccination status.

Comments:

As the authors stated in the discussion ‘the routine use of HL screening tools remains nevertheless controversial, as routine screening has shown no benefits yet could have undesirable effects.’ What message can clinician take home from this study?

Thank you. We believe that studies such as ours are useful to determine whether health literacy levels within a given population are adequate or inadequate, and whether such levels are associated to poorer clinical outcomes. Such learning would encourage clinicians and service providers to implement remedial interventions, including the use of ‘health literacy universal precautions’ that ensure all patients can benefit from information that is understandable enough. However, it is important to remember that we report here on a research experience and do not advocate the use of routine HL screening on all individual patients, for the reasons mentioned already. Further, our results suggest that inaction from healthcare workers regarding providing information and encouraging vaccination is responsible for most decisions by women to reject vaccination. Additionally, our results indicate that some women with high health literacy reject vaccination and clinician would therefore have to exercise greater skills to understand the decision-making frameworks in place for these women.

26) Although the vaccination rate for pertussis vaccine is high (94%), it will be good to collect and present data on important factors like 'information from health professionals on the pertussis vaccine' or media report, which may be the main factor that affect the vaccination rate between the influenza vaccination and pertussis vaccination rather than health literacy.

It would seem plausible that, if healthcare professional recommendations for influenza immunization were not effective to induce influenza vaccination decisions, then they would also be unlikely to have any effect on pertussis vaccination. We included a reference in our Discussion regarding the impact of news and media on vaccination with pertussis. We however did not suggest that health literacy was an explanatory factor regarding the difference in vaccination rates seen between the 2 types of immunisations.

Comments:

Could the authors provide plausible explanations regarding the difference in vaccination rates seen between the 2 types of immunisations?

Many thanks for encouraging us to explore this area. We have now included a few additional sentences in the Discussion, essentially suggesting that at least in our context, fear among women the surge in pertussis cases including infant deaths and which was extensively covered in local and national media may have played a key role in encouraging women to accept the pertussis vaccination. The fact that such surge led to the initiation of a national immunization programme would have reinforced women's' perspectives about the severity of whooping cough as a pathogen. Additionally, clinicians may have been more assertive recommending or encouraging women to have such vaccination, as a result of such media coverage. On the contrary, influenza disease and immunisation may have suffered from 'attention fatigue' so women and clinicians may have not afforded them sufficient consideration. Clearly, these suggestions would benefit from specific studies, most likely qualitative, exploring the impact of these perceptions on vaccine decisions and hesitancy adequately, as seen in other settings and already reported (Vilca LM, Martínez C, Burballa M, Campins M. Maternal Care Providers' Barriers Regarding Influenza and Pertussis Vaccination During Pregnancy in Catalonia, Spain. *Matern Child Health J.* 2018 Feb 7. doi: 10.1007/s10995-018-2481-6].

28. In page 7 the authors stated that 'Quantitative scores classify individuals with "adequate" (score: 37-50 points) or "inadequate" HL (score: 0-37 points).' Which group should a woman be if she got a score of 37, adequate or inadequate? I think the way reported in table 1 is a right way to do (adequate 38-50).

We are grateful for the precision. We have clarified and rectified appropriately now. Adequate scores (38-50).

FORMATTING AMENDMENTS (if any)

Required amendments will be listed here; please include these changes in your revised version:

- Kindly re-upload FIGURES with at least 300 dpi resolution.

Thank you, we have reformatted the figures to the appropriate resolution now.